# Prognostic Gene Predictors of Gestational Diabetes in Endometrium and Follicular Fluid of Women after Infertility

**DOI:** 10.3390/medicina58040498

**Published:** 2022-03-30

**Authors:** Brigita Vaigauskaitė, Raminta Baušytė, Elvina Valatkaitė, Giedrė Skliutė, Edita Kazėnaitė, Diana Ramašauskaitė, Rūta Navakauskienė

**Affiliations:** 1Department of Molecular Cell Biology, Institute of Biochemistry, Life Sciences Center, Vilnius University, Saulėtekio Av. 7, LT-10257 Vilnius, Lithuania; raminta.bausyte@santa.lt (R.B.); elvina.valatkaite@gmc.vu.lt (E.V.); giedre.skliute@bchi.stud.vu.lt (G.S.); ruta.navakauskiene@bchi.vu.lt (R.N.); 2Centre of Obstetrics and Gynaecology of the Institute of Clinical Medicine, Faculty of Medicine, Vilnius University, Santariškių St, LT-08661 Vilnius, Lithuania; diana.ramasauskaite@santa.lt; 3Faculty of Medicine, Vilnius University Hospital Santaros Klinikos, Vilnius University, Santariškių St, LT-08661 Vilnius, Lithuania; edita.kazenaite@santa.lt

**Keywords:** gestational diabetes mellitus, infertility, gene expression

## Abstract

*Background and objectives.* Gestational diabetes mellitus is an increasingly diagnosed metabolic disorder during pregnancy with unknown pathological pathways. Taking into account the growing numbers of women who are conceiving after assisted reproductive technologies, they comprise an engaging target group for gestational diabetes mellitus etiopathogenesis research. In terms of metabolism and genetics, as the evidence shows, both unexplained infertility and gestational diabetes mellitus pose challenges for their interpretation due to the complex bodily processes. *Materials and Methods.* Our study examined the expression of genes (*IGF2, GRB10, CRTC2, HMGA2, ESR1, DLK1, SLC6A15, GPT2, PLAGL1*) associated with glucose metabolism in unexplained infertility patients who conceived after in vitro fertilization procedure, were diagnosed with GDM and their findings were compared with control population. *Results*. There were no significant differences in gene expression of endometrium stromal cells between healthy pregnant women and women with gestational diabetes, although the significant downregulation of *CRTC2* was observed in the follicular fluid of women with gestational diabetes mellitus. Moreover, expression of *HMGA2* and *ESR1* was significantly reduced in FF cells when compared to endometrial cells. *Conclusions.* These findings may indicate about the importance of follicular fluid as an indicator for gestational diabetes and should be explored more by further research.

## 1. Introduction

Gestational diabetes mellitus (GDM) is a metabolic disorder defined as glucose intolerance (hyperglycemia) determined during pregnancy. According to literature, the prevalence of GDM is increasing and estimates around 5.8% of pregnancies in Europe, up to 7% in the United States and can reach 11.7% in Southeast Asia [1]. In Lithuania, the estimated GDM rate in 2019 was 12.4% [2]. The increase of this condition is associated with overweight and obesity prior to pregnancy, gestational weight gain from conception to 15–20 weeks of gestation and in older age [3,4]. These risk factors are well known and have a strong correlation with GDM, however do not necessarily predict it as there are many cases with no prior history of metabolic disturbances. Diagnosing GDM also puts the mother at risk of metabolic syndrome and type 2 diabetes mellitus [5], poses a higher risk of cardiovascular events [6]. In addition, GDM diagnosis can significantly worsen the quality of life as these women report a poor physical function and worse self-rated health status [7]. Along with these consequences observed in mothers, GDM makes a huge impact on fetal development. Affected fetuses are at a greater risk of macrosomia (which can lead to shoulder dystocia and birth complications); fetal hypoglycemia; an increased risk of respiratory distress, etc. [8]. Nowadays, more attention has been brought to GDM in pregnancy with history of infertility. Infertility is defined as a state when the couple is attempting to achieve a pregnancy for at least 12 months without conception. Not only women with prior ovulation disorders were at 52% greater risk of GDM, but also those diagnosed with unexplained infertility were exposed to a higher risk of GDM [9,10]. Moreover, some studies suggest that assisted reproductive technologies (ART) are associated with an increased risk of GDM [11,12]. Since infertility can affect up to 15% of couples of reproductive age [13], we believe that women with ovulation problems and unexplained infertility may be a target group for research aimed to understand the pathogenesis of GDM and its effects on both the mother and the fetus.

A wide scope of studies have made an attempt to better understand the etiopathogenesis of GDM. Obviously, there are around 1579 genes expressed by placenta related to GDM and its pathogenesis (such as LEP, MIF, FLT1, and UTS2) [14]. Some studies show that in pregnant women with GDM, a different DNA methylation profile is found, compared to non-GDM pregnancies [15]. Also, there are several studies which have examined associations between genetic variants and the risk of GDM and found significant found a significant association between the presence of this polymorphism and the risk of GDM [16,17,18]. However, there is a lack of data of the gene expression in a preconception period. Our study investigated nine genes (*IGF2, GRB10, CRTC2, HMGA2, ESR1, DLK1, SLC6A15, GPT2, PLAGL1*) affiliated to glucose metabolism, synthesis and metabolism. Since it has never been studied in GDM patients, we believe it can give crucial information in understanding GDM pathogenesis. In addition, there is a lack of information about how these genes are expressed in follicular fluid and endometrium, so our data could significantly supplement already existing studies on this topic. We did not yet investigate other gene expression (TNFα, PPARα or PPARγ) but it is our goal to extend the study in find gene expression in other tissues (for example, placenta). Current set of known genes is limited and certainly does not represent all potential gene features, but it would provide some crucial information of GDM pathogenesis. In the current study, we analysed genes associated with glucose metabolism and the observed differences between the endometrium and follicular fluid (FF). In our belief, FF could be the target media for further gene expression associations with gestational diabetes research. It will not only facilitate understanding the etiology of GDM, but will also provide recommendations for the patients on ongoing pregnancy care and health. 

## 2. Results

In the present study we analyzed changes in gene expression levels by comparing the results between healthy and gestational diabetes patients. The study we conducted represents nine genes (*IGF2, GRB10, CRTC2, HMGA2, ESR1, DLK1, SLC6A15, GPT2, PLAGL1*) associated with glucose metabolism, regulation of growth, gluconeogenesis and amino acid metabolism. First, we analyzed literature and gathered the most significant information about these genes. Then, we proceeded to perform gene expression analysis and finally, by combining both experimental data as well as information from databases, we designed an interaction network based on these nine analysed genes.

### Expression of Genes Inherent in Glucose Metabolism in Healthy and GD Patients

In the present study we analysed nine genes (*IGF2, GRB10, CRTC2, HMGA2, ESR1, DLK1, SLC6A15, GPT2, PLAGL1*) associated with glucose metabolism, regulation of growth, gluconeogenesis and amino acid metabolism. The genes, differentially expressed in endometrium and follicular fluid cells, and their functions are listed in Table 1**.**

By conducting RT-qPCR analysis, we evaluated a relative gene expression of *IGF2, GRB10, CRTC2, HMGA2, ESR1, DLK1, SLC6A15, GPT2, PLAGL1* genes (Figure 1 and Figure 2). Raw CT values for the investigated genes were included in the Appendix A (Appendix A). Endometrial stem cells and follicular fluid were isolated from healthy volunteers with normal glucose tolerance and from volunteers experiencing glucose intolerance—gestational diabetes patients. By performing RT-qPCR analysis, we determined that *IGF2* and GRB10 gene expression profiles demonstrate similar tendencies: the expression levels of these genes in endometrium stem cells were similar in both healthy and GD patients and was slightly upregulated compared to follicular fluid healthy and GD volunteers (Figure 1). The expression profile of *PLAGL1* indicates that this gene in endometrium and follicular fluid is not expressed actively since in endometrium the value of the relative expression is equal to zero and in FF it is barely increased. On the contrary, the expression of *DLK1* gene in the follicular fluid’s group of healthy and GD volunteers was increased notably in comparison to the Endo group which was maintained at similar, but low levels (Figure 1). 

Next, our study found that relative *ESR1* and *HMGA2* gene expression in endometrium stem cells was similar and did not differ significantly between healthy and GD patient groups, however, it was slightly higher compared to the follicular fluid group (Figure 2). Similarly, to the Endo group, follicular fluid expressed *ESR1* gene in healthy and GD patients at an almost identical level, although the observed expression was at very low levels. Additionally, we detected that *CRTC2* gene’s relative expression values in endometrium stem cells of healthy and GD patients were at a comparable level to follicular fluid’s healthy and GD patients (Figure 2). Also, the expression of this gene in the endometrium stem cells of both volunteer groups did not display significant differences, however, a few healthy volunteers in FF group did exhibit an increased activity of *CRTC2* gene.

*SLC6A15* and *GPT2* genes show similar changes in their gene expression profiles (Figure 2). The expression of *SLC6A15* in the endometrium stem cells of healthy and GD patients was very low and similar to that of *GPT2* gene. Moreover, the expression of *SLC6A15* in FF group‘s patients was limited and did not display any significant changes in the expression levels between the two patient groups, which is also seen in the *GPT2* gene expression profile. Overall, the gene expression profiles presented in this study demonstrated that in most cases these 9 genes did not display any significant differences between healthy patients and patients with gestational diabetes. The only observed differences were significant downregulation of CRTC2 gene in follicular fluid of patients with GDM and changes of gene expression levels between the endometrium and follicular fluid.

Below we have presented an interaction network between *IGF2, GRB10, CRTC2, HMGA2, ESR1, DLK1, SLC6A15, GPT2, PLAGL1* genes (Figure 3). We analyzed how these genes interact with each other in the context of gestational diabetes as well as glucose metabolism and an overall involvement in the occurrence of diabetes. From the network we can see that all of the genes are connected with one another with the exception of *GPT2, CRTC2* and *SLC6A15.* For example, it was shown that imprinted genes *IGF2, PLAGL1, DLK1* and *GRB10* are downregulated in multiple organs with age. The authors suggested that these genes are upregulated in late embryonic and early postnatal life, but later are decreased. This shows that these four together with other genes coordinate deceleration in multiple organs and tissues and contribute to cell proliferation [19].

## 3. Discussion

Gestational diabetes is commonly described as any degree of hyperglycemia occurring at the onset of pregnancy. GD in pregnant women represents a broad spectrum of hyperglycemia, including mild glucose intolerance, which is usually detected in late pregnancy and glucose levels indicative of diabetes, which is detected in early pregnancy [20]. It has become a more prevalent problem in recent years due to an increase in diabetes and obesity cases worldwide [21,22]. Despite the advanced medical diagnostic tools and treatments, gestational diabetes still poses a risk not only to a mother, but to a fetus as well. Some adverse outcomes affecting maternal side of pregnancy include pre-eclampsia, preterm labour, nephropathy, birth trauma, cesarean section, and postoperative wound complications. Fetal complications associated with GD include altered birth weight or excess fetal adiposity, congenital anomalies, shoulder dystocia, stillbirth, and hypoglycemia [23]. 

FF is secreted by granulosa cells in the follicles [24]. Follicular fluid research has gained broader attention of scientists since its composition and importance for the oocyte development has been estimated [25]. Not only it nourishes the oocytes and is filled with proteins, hormones, fatty acids, inflammatory factors, etc., but also cell-free DNA has been found therein which negatively correlates with follicle size and embryo quality [26]. Likewise, Qing Sang et al. investigated miRNA in FF and showed that there are numerous miRNA which take part in steroidogenesis and are significantly associated with the polycystic ovarian syndrome (PCOS) [27]. Further, some animal studies investigated glucose concentration in FF and found that glucose concentration changes in FF affect oocyte maturation and growth of granulosa and cumulus cells affecting directly the oocyte’s competence [28]. These studies imply that FF brings us important information not only about the oocyte, but also embryo and its future development. Similarly to the way how miRNA predicts PCOS, we believe gene expression in FF may affect the embryo‘s development and have an impact on the ongoing pregnancy.

The endometrium is a renewable tissue and its cells play a particularly important role during a woman’s reproductive life. More than a few hundred regenerative processes are believed to occur in this tissue. Endometrial stromal cells are thought to be crucial for endometrial physiology, pathology, embryo implantation and development. Since endometrium is such an important medium for the embryo, an increasingly accelerating number of studies have been conducted to investigate gene expression in the endometrium in order to understand its ability to accept the embryo for implantation. D. Haouzi et al. for the first time found that the genes (MFAP5, ANGPTL1, EG-VEGF, NLF2) being up-regulated during the implantation window, could be predictive for endometrial receptiveness [29]. Hypothesis suggests that EG-VEGF (the endocrine gland-derived vascular endothelial growth factor) may regulate proliferation, angiogenesis and permeability, and induce the formation of endothelial fenestration. Other studies examine glucose metabolism in endometrium and its potential effect on pregnancy. Glucose transporters (GLUT1, GLUT3, GLUT4, GLUT8, SGLT1GLUT3) are expressed in endometrium and play an important role in the differentiation and decidualisation of the endometrium, embryo implantation, glucose uptake in the first trimester of pregnancy although their full mechanism of action is still unclear [30]. Since GLUT3 is a high-affinity glucose transporter expressed in the proliferative phase and decidua at a constant level throughout different menstrual stages or in early pregnancy, it is assumed that endometrium could be in charge of metabolism diseases in pregnancy [31].

In this study, we chose to explore gestational diabetes on a molecular level by studying *IGF2, GRB10, CRTC2, HMGA2, ESR1, DLK1, SLC6A15, PLAGL1* and *GPT2* genes associated with glucose metabolism, growth regulation and gluconeogenesis. Our study based on the performance of the RT-qPCR analysis demonstrated that IGF2 and GRB10 genes display similar expression levels, where in endometrium stem cells the expression of these genes was upregulated compared to follicular fluid. Interestingly, the patients of both groups exhibited almost identical expression levels, meaning that IGF2 and GRB10 genes are maintained at similar levels in healthy and in gestational diabetes patients. It is known that IGF2 is a member of insulin-like growth factor signaling pathway, which is primarily responsible for the regulation of the size of the fetus and placenta during gestation. A study conducted by Su and colleagues demonstrated that the expression of insulin-like growth factor 2 is upregulated significantly in GD group compared to a normal glucose tolerance group [32]. The samples were taken from placenta and umbilical cord, and IGF2 expression was measured using RT-qPCR. Another study proved the correlation between an elevated IGF2 expression and an increased fetus organ size [33]. Another imprinted gene GRB10 coordinates the fetal and placental growth and regulates glucose and lipid metabolism, although its regulatory role in growth is independent from the insulin-like growth factor signaling pathway [34,35,36]. More research studies suggest that the exposure of the fetus to intrauterine hyperglycemia accelerates the risk of abnormal glucose metabolism, obesity or hypertension in these infants in their adulthood [32]. It was previously demonstrated that transient neonatal diabetes mellitus is associated with overexpression of PLAGL1 gene, although our study did not reveal many changes in GD patients [37]. The next gene we focused on was DLK1 which takes part in glucose metabolism and which definitely displayed significant differences between endometrium stem cells and follicular fluid. We determined that the relative DLK1 expression in FF group was highly upregulated compared to endometrium stem cells, however, expression levels in healthy and GD patients were very similar and did not indicate significant changes. Similar observations were published in Wurst et al. article, on the examination of DLK1 levels in subjects with gestational diabetes and healthy patients, however, no significant differences among these two groups were found [38]. ESR1 was the following gene under our analysis. ESR1 is a transcription factor that encodes estrogen receptor α (ER α) which is involved in the regulation of growth, metabolism, energy balance, gestation and sexual development. Our present study demonstrates that ESR1 gene is not expressed actively in the endometrium or in the follicular fluid in GD patients compared to a healthy control group. Similar results were obtained by Kleiblova et al. who collected placentas from 23 pregnant women and did not find significant differences in ESR1 gene expression levels between the GD group and the control group [39]. On the contrary, Knabl et al. (2015) claim that the ESR1 gene expression levels in decidua tissue containing extravillous trophoblasts and decidua stromal cells were upregulated in GD patients compared to normal controls [40]. The study we have conducted also involved the evaluation of the HMGA2, CRTC2 and SLC6A15 gene expression levels, however, in most cases healthy and GD patients expressed similar levels of these genes and did not yield to significant changes. Unfortunately, there are scarce publications on linking HMGA2, CRTC2 and SLC6A15 genes to gestational diabetes, so further research on this topic needs to be carried out to enable the comparison of our findings presented in this study with other data. Finally, we evaluated the GPT2 expression levels which were similar to the profile of the PLAGL1 gene—healthy and GD patients expressed similar levels of this gene in both, the Endo and FF groups. An interesting study was conducted by Jungheim et al. (2011), where murine blastocysts were cultured in obesity inducing medium, containing excess palmitic acid (PA) and glutamic pyruvate transaminase (GPT2) activity was measured [41]. The authors found that PA exposure on murine embryos resulted in altered IGF1R activity and increased GPT2 activity compared to the embryos cultured in the control medium. The increased GPT2 activity was suggested to be involved in abnormal insulin signaling.

The results of this research clearly showed that further studies are necessary in order to conclude a causal association between the glucose gene expression in FF and the prevalence of GDM in the infertile women population. Not only the number of participants is therefore considered quite small to gain a variety of perspectives but also gene expression could vary within and between populations. Although, we believe that the design of this study is novel and will help to guide the future studies to new prospective.

## 4. Materials and Methods

### 4.1. Study Population and Data Collection

We have conducted an interventional prospective cohort study. The patients were enrolled from 1 January 2018 to 30 March 2020 at the Vilnius University Hospital Santaros Klinikos Obstetrics and Gynecology Center who were treated in Santaros Fertility Center and gave birth in the department of Obstetrics. Females with unexplained infertility who received ART and later were diagnosed with GDM were enrolled in this study. Although, 13.3% of patients received insulin treatment, the rest of the patients were treated with medical nutrition therapy, together with weight management and physical activity. The inclusion criteria were as follows: (1) the age of women at the time of enrollment was 25–35 years, (2) the average duration of infertility at least 2 years, (3) the unexplained infertility diagnosis was confirmed after laboratory and instrumental investigation, (4) the women who conceived after ART, (5) the women who were diagnosed with GDM during pregnancy, by taking a 3-h oral glucose tolerance test, (6) informed consent of all the subjects was received. The exclusion criteria were as follows: (1) women had prior diagnosis of diabetes mellitus, (2) oncological disease was confirmed for woman during the last three years, (3) other infertility causes, except unexplained infertility, were confirmed, (4) women who were addicted to alcohol or other substances, (5) uncontrolled endocrine or other medical conditions, such as prolactinemia or thyroid diseases. The characteristics of the study population is presented in Table 2.

### 4.2. Isolation and Cultivation of Human Endometrium Derived Mesenchymal Stem Cells and Follicular Fluid

In this study protocols and experimentations with the patients’ specimens were approved by the Ethics Committee of Biomedical Research of Vilnius District, No. 158200-18/7-1049-550. Endometrium stem cells and follicular fluid were obtained from patients undergoing ART at Vilnius University Hospital Santaros Fertility Center. The specimens were isolated from a total of 26 healthy and GDM donors. 13 patients were selected for the collection of endometrial samples and 13 patients for the collection of FF. Both endometrium and follicular fluid were obtained from women diagnosed with unexplained infertility undergoing in vitro fertilization (IVF) procedure. All women were treated with antagonist protocol (recombinant follicle-stimulating hormone together with gonadotropin-releasing hormone antagonist) to achieve a controlled ovarian stimulation. 

Endometrium samples were collected during the scratching procedure without dilatation of the cervix, using a pipelle catheter during the natural luteal cycle phase (approximately 7 days before menstrual cycle and start of ovarian stimulation). Endometrium stem cells were isolated as described previously [42] by performing endometrial biopsy on the inner wall of endometrial lining. Isolated cells were then seeded into 25 cm^2^ plastic cell culture flasks at 1.6 × 10^4^ cells/cm^2^ density and maintained at 37 °C, in a humidified 5% CO_2_ incubator. The cells were cultured in a growth medium DMEM/F12 (DMEM, Dulbecco’s Modified Eagle Medium/Nutrient Mixture F-12) (Gibco, Thermo Fisher Scientific, Waltham, MA, USA) supplemented with 10 % Fetal Bovine Serum (FBS) (Gibco, Thermo Fisher Scientific, Waltham, MA, USA), 100 U/mL penicillin and 100 µg/mL streptomycin (Gibco, Thermo Fisher Scientific, Waltham, MA, USA), which was replaced every 2–3 days. 

Follicular fluid containing heterogenic population was collected at the time of oocyte aspiration without flushing (36 h after human chorionic gonadotropin (6500 IU) trigger administration) into sterile tubes. Follicular fluid samples used for analysis were macroscopically clear and not contaminated with blood. Once received in the laboratory, samples were transferred into 50 mL tubes and centrifuged at 500× *g* for 10 min at RT, then supernatant was removed and 20 mL of 1× red blood cell (RBC) lysis buffer (10×, 155 mM NH_4_Cl (Sigma-Aldrich, St. Louis, MO, USA), 12 mM NaHCO3 (Sigma-Aldrich, St. Louis, MO, USA), 0.1 mM EDTA (Sigma-Aldrich, St. Louis, MO, USA), pH 7.3) added to the pellet, incubated for 5 min at RT and centrifuged at 500× *g* for 5 min at RT. The procedure was repeated if there were erythrocytes remaining. After isolation cells were washed in phosphate-buffered saline (PBS) (Gibco, Thermo Fisher Scientific, Waltham, MA, USA) twice and counted.

### 4.3. RNA Isolation and Gene Expression Analysis Using RT-qPCR

Total RNA was extracted from endometrium stem cells and 1 × 10^6^ mononuclear cells using TRIzol^®^ reagent (Zymo Research, Irvine, CA, USA) and Quick-DNA/RNA™ Miniprep Kit (Zymo research, Irvine, CA, USA) respectively according to manufacturer’s instructions. cDNA was synthesized with LunaScript^®^ RT SuperMix Kit (New England Biolabs, Ipswich, MA, USA) following manufacturers recommendations. RT-qPCR was performed using Luna^®^ Universal qPCR Master Mix (New England Biolabs, Ipswich, MA, USA) and Rotor-Gene 6000 Real-time Analyzer (Corbett Life Science, QIAGEN, Hilden, Germany) with experimental conditions as follows: 95 °C 1 min, 95 °C 15 s, 60 °C 30 s. The list of primers and their sequences used in gene expression analysis is presented in Table 3. The acquired data from RT-qPCR was further analysed, each sample’s mRNA levels were normalised to GAPDH gene expression and relative gene expression was calculated using ∆∆Ct method.

## 5. Conclusions

This study has for the first time investigated the gene expression associated with glucose metabolism in pre-conception women. It allowed us to take a look and search for gestational diabetes development at a very early stage. Although we detected only insignificant and slight differences in gene expression between healthy and gestational diabetes patient groups, we believe follicular fluid could be a target medium for further research of pregnancy pathologies.

## Figures and Tables

**Figure 1 medicina-58-00498-f001:**
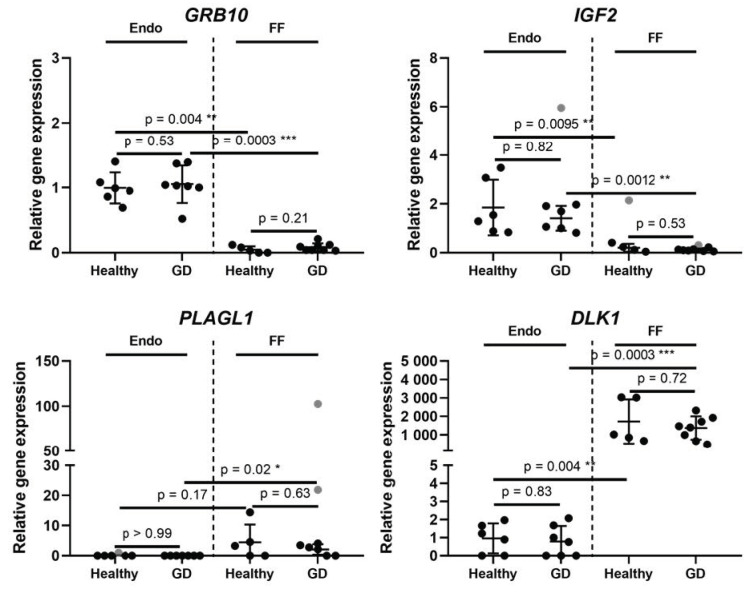
Expression of growth-regulating imprinted genes inherent in glucose metabolism and occurrence of diabetes. Results are presented as mean ± standard deviation, grey data points indicate outliers. Endo stands for endometrium, FF—follicular fluid, GD—Gestational diabetes. The Endo group’s healthy patients *n* = 6, the endo group’s GD patients *n* = 7. The FF group’s healthy patients *n* = 5, the FF group’s GD patients *n* = 8. Statistical significance is based on the Mann-Whitney test, * *p* ≤ 0.05; ** *p* ≤ 0.01; *** *p* ≤ 0.001; outliers were determined by ROUT (Q = 5%).

**Figure 2 medicina-58-00498-f002:**
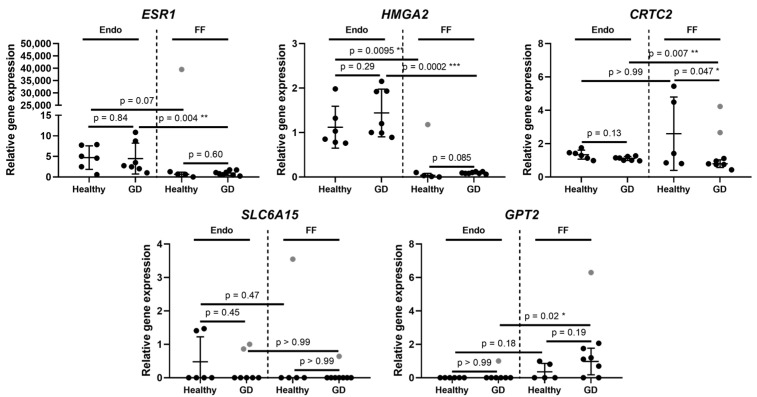
Expression of gluconeogenesis and amino acid metabolism related genes. Results are presented as mean ± standard deviation, grey data points indicate outliers. Endo stands for endometrium, FF—follicular fluid, GD—Gestational diabetes. The Endo group’s healthy patients *n* = 6, GD patients *n* = 7. The FF group’s healthy patients *n* = 5, GD patients *n* = 8. Statistical significance is based on the Mann-Whitney test, * *p* ≤ 0.05; ** *p* ≤ 0.01; *** *p* ≤ 0.001; outliers were determined by ROUT (Q = 5%).

**Figure 3 medicina-58-00498-f003:**
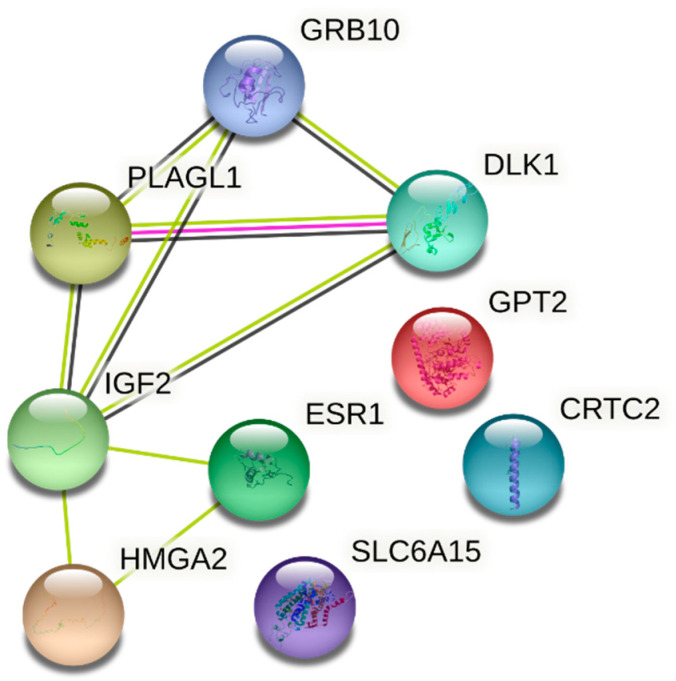
Interaction networks of the genes under analysis inherent in sugar metabolism and occurrence of gestational diabetes. The interaction networks were identified using the STRING database. Lines between nodes represent protein–protein associations. CRTC2 stands for CREB regulated transcription coactivator 2; DLK1—delta like non-canonical Notch ligand 1; ESR1—estrogen receptor 1; GPT2—glutamic--pyruvic transaminase 2; GRB10—growth factor receptor bound protein 10; HMGA2—high mobility group AT-hook 2; IGF2—insulin like growth factor 2; PLAGL1—PLAG1 like zinc finger 1; SLC6A15—solute carrier family 6 member 15. Green line represents co-mentioning in PubMed abstracts; black line—co-expression; pink line—experimental/biochemical data based on STRING analysis.

**Table 1 medicina-58-00498-t001:** List of differentially expressed genes in endometrium and follicular fluid cells of patients with gestational diabetes.

Gene Symbol	Gene Name	Gene Ontology (Molecular Function)	Location	Endometrium		Follicular Fluid	
				Fold	*p* *	Fold	*p* *
*CRTC2*	CREB regulated transcription coactivator 2	Transcription of genes targeted by the cAMP response element-binding protein	1q21.3	0.83	0.13	0.31	0.047
*DLK1*	Delta like non-canonical Notch ligand 1	Regulation of cell growth and differentiation	14q32.2	0.82	0.83	0.79	0.72
*ESR1*	Estrogen receptor 1	Transcription factor	6q25.1-q25.2	0.95	0.84	1.47	0.60
*GPT2*	Glutamic--pyruvic transaminase 2	Catalysation of the reversible transamination. Play roles in gluconeogenesis and amino acid metabolism	16q11.2	0	>0.99	2.71	0.19
*GRB10*	Growth factor receptor bound protein 10	Interaction with insulin receptors and insulin-like growth-factor receptors	7p12.1	1.06	0.53	1.82	0.21
*HMGA2*	High mobility group AT-hook 2	Component of the enhancesome. Transcriptional regulating factor, enhancing or suppressing the transcriptional activity depending on the number and spacing of the AT-rich binding sites in DNA	12q14.3	1.29	0.29	2.61	0.085
*IGF2*	Insulin like growth factor 2	Growth factor, involved in development and growth	11p15.5	0.76	0.82	0.57	0.53
*PLAGL1*	PLAG1 like zinc finger 1	Suppressor of cell growth. Overexpression of this gene during fetal development is thought to be the causal factor for transient neonatal diabetes mellitus	6q24.2	0	>0.99	0.47	0.63
*SLC6A15*	Solute carrier family 6 member 15	Transport of neutral amino acids	12q21.31	0.45	0.45	0	>0.99

cAMP, cyclic adenosine monophosphate; Fold, fold difference. Fold difference is calculated comparing GD group to control group. * *p* value: Statistical significance is based on Mann-Whitney test.

**Table 2 medicina-58-00498-t002:** Characteristics of the study population.

Characteristics	GDM Cases (*n* = 15)	Controls (*n* = 12)
Maternal age, in years	33.8 ± 3.6	31.7± 2.9
BMI before pregnancy	23.4 ± 4.1	22.1 ± 3.8
GDM insulin treatment, %	13.3	-
Gestational age at delivery, wks.	37 ± 2.7	38 ± 4.2
Delivery mode, % Vaginal Cesarean	80.0 20.0	83.3 16.7
Preterm rupture of membranes, %	33.3	41.6
Nulliparous, %	86.6	33.3
Pregestational obesity, %	13.3	16.6
Macrosomia, %	20.0	25.0

**Table 3 medicina-58-00498-t003:** List of primers used in RT-qPCR for gene expression analysis.

Gene Name	Primer Sequence (5′-3′), F—Forward, R—Reverse	Product Size (bp)
*GAPDH*	F: GTGAACCATGAGAAGTATGACAAC R: CATGAGTCCTTCCACGATACC	123
*IGF2*	F: TCGTTGAGGAGTGCTGTTTCC R: ACACGTCCCTCTCGGACTTG	87
*GRB10*	F: GCAAACAGGACGCGTGATAGAG R: GTGAATCACTGTACTTAGGG	148
*CRTC2*	F: CTCTGCCCAATGTTAACCAGAT R: GAGTGCTCCGAGATGAATCC	87
*HMGA2*	F: CCCAAAGGCAGCAAAAACAA R: GCCTCTTGGCCGTTTTTCTC	81
*ESR1*	F: TGGGAATGAAAGGTGGGATAC R: AGGTTGGCTCTCATGTC	124
*DLK1*	F: TCCTCAACAAGTGCGAGACC R: CTGTGGGAACGCTGCTTAGA	191
*SLC6A15*	F: TGCTAATAGCTAGTGTTGTGAATATGG R: GGATAGCTCAGAAATTCTTCAGATG	94
*GPT2*	F: GAACATAAGCGAATGAAGAAAGAAG R: CCAGCAGGTTTGGGTAGGT	294
*PLAGL1*	F: GGCGGGTAGGTAGGAAAGG R: GATGTTTATGAATCAGGCAGG	49

## Data Availability

The data presented in this study are available on request from the corresponding author.

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
