# Peer review of "Prognostic Gene Predictors of Gestational Diabetes in Endometrium and Follicular Fluid of Women after Infertility"

_medicina, 2022, doi:10.3390/medicina58040498_

Round 1
Reviewer 1 Report
Major comments:
The paper entitled “Prognostic Gene Predictors of Gestational Diabetes in Endometrium and Follicular Fluid of Women after Infertility” by Vaigauskaite et al, is collectively aimed at understanding the array of genes involved in glucose metabolism during gestational diabetes mellitus and the significance of follicular fluid in GDM research. However, the following points needs to be addressed.
- The rationale for selecting the 9 genes is to be explained. This is very important for readers to understand the manuscript fully.
- In table 1, the gene HMGA2 is mentioned as a transcriptional regulatory factor. To be more precise, it can be mentioned whether it belongs to transcriptional enhancer or repressor.
- In the abstract, it is mentioned only CRTC2 is significantly downregulated. However, Figure 2 indicates significant reduced level of HMGA2 and ESR1 when compared between endometrial cells and FF.
- The author’s needs to mention the limitations of computational prediction method followed in the manuscript
- The authors can provide amplification plot and dissociation curve of the gene expression data as supplementary data to verify the product specificity.
- The authors need to justify the selection of EC and FF to assess the gene expression.
- The treatment regime for GDM needs to be elaborated. For instance, 13.3 % of GDM population received insulin treatment. The treatment regime for the remaining percentages were not spelled out. It is necessary to understand the significant difference in the gene expression between control and GDM population. The oral hypoglycemic agents could interfere with the expression levels of these genes.
- The conclusion drawn in the abstract does not carry the word diabetes and is significantly different from the title of the manuscript.
- The authors should refrain from using the term sugar throughout the manuscript and replace sugar with glucose.
Minor comments:
- In abstract, line number 23, the term groups can be replaced with the term population, as population is the appropriate terminology to be used in any clinical studies.
- The term sugar in the manuscript must be replaced with glucose.
- It is necessary to expand the abbreviation while using for the first time in the manuscript. The abbreviation for follicular fluid (FF) is stated in different places in the manuscript and its expansion is given in different places. The uniformity of the abbreviation is important.
- The abbreviation for gestational diabetes mellitus (GDM) is stated in different places in the manuscript and its expansion is given in different places. The uniformity of the abbreviation is important.
- In line number 278, change “women were addicted” to “women who were addicted”.
Reviewer 2 Report
Comments:
- The paper does not appear to have a methods section or the order needs to be adjusted
- ; could the authors please provide this please?
- Could the authors please provide ethical approval for their study please
- Could the authors clarify the timing of endometrial biopsies with respect to the genes studied please?
- Could the authors please justify the number of replicates for each experiment please?
- It appears that the authors have used parametric statistical analyses; could the authors please justify this approach please?
Round 2
Reviewer 1 Report
The authors have adequately addressed the queries on the previous version of the manuscript, therefore the manuscript may be accepted for publication
Reviewer 2 Report
The authors have satisfactorily responded to my comments.